# Gender difference on the mediation effects of filial piety on the association between chronic obstructive pulmonary disease and depressive symptoms in older adults: A community-based study

**Cheng-Chen Chang[1,2], Kaichi Hsu[3], Chun-Min Chen[4], Si-Sheng Huang[1], I-Chien Wu[3], Chih-Cheng Hsu[3], Agnes C. Hsiung[3], Hsing-Yi Chang[3] ***

1 Department of Psychiatry, Chung Shan Medical University Hospital, Taichung, Taiwan, 2 School of Medicine, Chung Shan Medical University, Taichung, Taiwan, 3 Institute of Population Health Sciences, National Health Research Institutes, Maoli, Taiwan, 4 Big Data Center, Changhua Christian Hospital, Changhua, Taiwan

\* hsingyi@nhri.edu.tw

**Data Availability Statement:** The data is not available to the public currently, due ethical

## Abstract

Filial piety is viewed as strong family support for older Chinese people, and strongly associated with depressive symptoms. It is unknown if there exists gender difference in the mediation effects of filial piety on the relationship between chronic obstructive pulmonary disease (manifested as lung function) and depression. We investigated whether filial piety mediates the association between lung function and depression in community-dwelling older men and women using the Healthy Aging Longitudinal Study in Taiwan (HALST). Community dwelling adults aged 65 and above were analyzed. Pulmonary function, depressive symptoms, and filial piety expectation (FPE) and receipt of filial piety (RFP) were collected. The interaction and mediation of filial piety between lung function and depression was analyzed. We found that in older men, forced expiratory volume in the first second (FEV1) was inversely correlated with depression ($\beta = -0.1281$, $p = 0.004$) with no mediation effect of FPE. In older women, FEV1 was negatively associated with FPE, but FPE did not increase the risk of depression ($\beta = 0.0605$, $p = 0.12$). In both older men and women, FEV1 was negatively associated with RFP, while RFP reduced the risk of depression ($p < 0.001$). In older women, the correlation between FEV1 was complete mediation of RFP. Results indicate that feelings of insufficient filial piety may increase the likelihood of depression, especially in elderly women with worse lung function. Although modest, the main mediation effect of filial piety was improvement of lung function in older subjects, which might decrease depression.

## Introduction

Filial piety is a basic virtue and traditional cultural value in Asia, and is viewed as a strong family support for older people. In a meta-analysis, the absence of filial piety was correlated with

restrictions on sharing data publicly. The data is owned by the National Health Research Institutes. It is only available to the investigators in the study project. Other researchers could propose for data analysis through collaboration with the investigators. The data can only be analyzed in the National Health Research Institutes. The data can be requested through our ethical committee at nirg@nhri.edu.tw.

**Funding:** Part of this study was sponsored by the National Health Research Institutes (PH-110-SP-01) and Changhua Christian Hospital (CCH grant no. 105-CCH-IRP-001). There was no additional external funding received for this study. The funders had no role in study design, analysis, decision to publish, and preparation of the manuscript.

**Competing interests:** The authors have declared that no competing interests exist.

depression in older people [1]. Filial piety can be viewed as part of social support from family. Social support might protect individual from depression. A meta-analysis reviewed 33 western studies examining social support and protection against depression in older adults (50 years or older), and found 90% of them reported significant positive protection from depression in some aspects of social support [2]. Similar study was done in 24 Asia population [3]. The study found association between good social support and decrease depression among older adults. Family support was found to have greater influence on depression in Asia than in the western population. The gender-specific analysis showed that slightly more women than men relied on social support [4].

Chronic obstructive pulmonary disease (COPD) is a major cause of chronic morbidity and mortality globally. According to the Global Burden of Disease, COPD is the third leading cause of death worldwide [5], accounting for 3.17 million deaths in 2015 (5% of all deaths globally that year) [6]. In 15 years, COPD is projected to be the leading cause of death and moderate to severe disability among the elderly. According to the Rotterdam study, the incidence of COPD has increased by two-fold from the ages of 55–59 to 75–79 years [7]. With the growing aging population in coming decades, the prevalence of COPD in older adults is expected to increase substantially, and will take a toll on health care and long-term care systems.

An association between COPD and depression is well recognized. Empirical studies have shown that depressive disorders are common in patients with COPD [8, 9]. The prevalence of depression in outpatients with COPD is between 13% and 46% [10, 11]. In a systemic review and meta-analysis, the pooled odds ratio of depression was three-times higher in stable COPD patients than in non-COPD controls [12]. Although the precise prevalence rate of depression in older adults with COPD is unknown, estimates range from 7% to 46% [8, 13, 14]. This wide discrepancy may be due to age, socioeconomic status, or diagnostic criteria for depression. However, the mechanisms that lead to depression in patients with COPD are complex and poorly understood. Although some studies have attempted to identify the biological determinants of depression in COPD patients, many socio-psychological aspects of depression in COPD patients remain unknown [10]. Previous studies have reported that experiencing stressful life events is significantly associated with the severity of COPD [15, 16], but the study samples have been relatively small. Symptoms of anxiety and depression are more common in female patients with COPD [17], and gender differences in the manifestation and treatment of COPD and depression have been reported [17, 18].

The mediating effect of social support on the relationship between frailty and depression was observed [19]. Study also found that extra-family support mediated the association between extraversion and depressive symptoms [20]. There is a strong association between COPD and depressive symptoms; however, the mediating effect of filial piety, a form of social support, on this association is unknown. Since deterioration of lung function can be an indicator of COPD and there is gender difference in the effects of social support, we hypothesized that lung function is inversely related to depression, and the relationship is mediated by filial piety differently in different gender. We used the Healthy Aging Longitudinal Study in Taiwan (HALST) study to explore our hypotheses.

## Material and methods

This is a community-based study. Data for this study were from the HALST [21], which is a longitudinal study recruiting people aged 55 and above from seven selected communities in Taiwan. A total of 5664 participants were enrolled in the first wave of the study since October 2008. Residents aged 55 years or above were recruited from the neighborhood of 7 hospitals located in the north, central, south, and east of Taiwan. This study was approved by the

internal review boards of the National Health Research Institute and participating hospitals (approval code: EC1020805, EC0970608, EC1020805, EC1090903, 105-CCH-IRP-001). All participants were informed about the study and written consent forms were obtained. Demographics, disease history, dietary pattern, cognitive function, quality of life etc. were asked. Fasting blood were drawn and sent to a certified clinical laboratory for analyses. Five percent repeated measures were taken. The inter- and intra-assay coefficients of variation (CV) were within reasonable range. Details were reported elsewhere [22]. The second wave started in 2013. Since pulmonary function was taken between 2016 and 2019, we analyzed data from the second wave with pulmonary function only. There were 1462 remained in the analysis.

## Pulmonary function tests

Participants were excluded if they had the following conditions within 2 months before pulmonary function tests: chest or abdominal surgery, sudden heart attack, hospitalization due to heart disease or respiratory infection, glaucoma, or cataracts. Pulmonary function was measured in the second wave of the HALST from 2016 to 2019. Subjects were asked to hold the spirometry using two hands while standing, and blow with maximum effort. The test was repeated five times. Forced vital capacity in liters (FVC [L]), forced expiratory volume in 1 s (FEV1 [L]), percent of FEV1/FVC (FEV%), forced exploratory volume in the 6th second (FEV6 [L]), peak expiratory time (PEF [L/min]), forced expiratory time (FET [s]), and 25–75% of peak expiratory flow time (F2575 [L/s]) were recorded. Only FEV1 and FVC were analyzed in this study. The average of five repeats was used in the analyses. COPD was defined as FEV1 < 70% predicted or FEV1/FVC < 0.7 [23].

## Filial piety

Filial piety was assessed in the Population Study of Chinese Elderly in Chicago (PINE) [24] and included two aspects: filial piety expectation (FPE) and PRFP. The expectation and actual experiences of six domains of filial piety including respect, happiness, care, greetings, obedience, and financial support were asked. These six domains were constructed by Gallois and colleagues [25] based on a conceptual model. A five-point scale was implemented for the participants to evaluate the six domains they expected from their children (from 1 = very little to 5 = very much) [26]. The overall FPE was calculated by adding each expectation score of the six filial behaviors. The overall expectation ranged from 6 to 30, with a higher score indicating a higher level of FPE. Similarly, the participants were asked to evaluate their receipt of care, respect, greeting, happiness, obedience, and financial support. The overall PRFP was calculated and the aggregate score ranged from 6 to 30, with a higher score indicating a higher level of RFP.

## Center for epidemiologic studies depression scale questionnaire

Depression symptoms were measured using a shorter version of the Center for Epidemiologic Studies Depression Scale (CES-D), which has been validated in different ethnicities [27] and contains 10 items. Each item was answered on a scale of 0 (rarely or none of the time) to 3 (most or almost all the time). The sum of the scores was analyzed.

## Measurement of other relevant covariates

Demographic characteristics, weight and height, blood profiles, and information on health-related behaviors and living arrangement were also obtained. Since some variables were significantly different in people with COPD from those without (S1 Table), age, gender, waist,

education, smoking status, hypertension, stroke, and MMSE were adjusted in the final model. Authors were allowed to analyze the data without real identification for individual.

## Statistical analyses

Descriptive statistics such as the mean ± standard deviation and percentage were used. Linear regression was used to investigate the relationship between lung function (measured by FEV1) and depressive symptom (measurement by CES-D). We hypothesized that there is a mediation effect on the relationship between FEV1 and CES-D. In doing so, the effects of confounders (FPE, PRFP) on the relationship between FEV1 and depression were assessed as follows: testing the direct predictive effect (c) of FEV1 on CES-D, testing the mediating effect, testing the interaction effect (a * b) of FEV1 on CES-D, and testing the indirect effect (c') of FEV1 on CES-D through the mediator (M). Of the associations among FEV1, depression, and FPE/PRFP, possible associated factors were also adjusted. Furthermore, because features of FEV1, FPE, and PRFP were significantly different by sex, we examined the moderating and mediating effects of FPE and PRFP between FEV1 and CES-D stratified by sex (Table 3). Statistical analyses were conducted using SAS version 9.4. Procedure CAUSALMED was used for mediation analysis. Sensitivity analysis was conducted using taking subsamples with age older than 65 (N = 1261) or with COPD (N = 362). Sensitivity analysis was conducted using subgroups of people, such as aged 65 and above and with COPD only.

## Results

There were 1462 subjects analyzed in this study. The characteristics of participants and number of missing values are presented in suppletory S1 Table. Since the proportion of missing was relatively small, we conducted the analyses directly. The features of FEV1, FPE, and PRFP as well as each component of FPE and PRFP in men and women are presented in Table 1. FEV1, FPE, and PRFP were significantly different between men and women (Table 1). Therefore, in the second step of the analysis, we examined the moderating and mediating effects of filial piety between FEV1 and CES-D in men and women, respectively.

 The direct effect between pulmonary function and depressive symptom was confirmed regarding the FEV1 on CES-D for men and women separately. The mediation effects of FPE and PEPR were also examined via regression of FPE (or PEPR) on CES-D. Then we tested the interaction effect of FPE and PRFP on the relationship between FEV1 and depression. In the model 1 analysis, FEV1 was entered as the independent variable; the CES-D scores were the dependent variables; FPE was the moderator variable if an interaction existed between FEV1 and MMSE; and age, waist circumference, education, smoking status, hypertension, stroke and MMSE, which were controlled in the mode. The results of models 1 and 2 revealed that the interactions were not significant (p > 0.05) and did not indicate a moderation effect of FPE or PRFP in both sexes (Table 2).

 The mediation pathways are depicted in Fig 1. Line c indicates the direct association c, and c' was the indirect relation after mediation (Fig 1). Statistical significance is presented as a black line, whereas the non-significance is depicted as a gray line. Table 3 shows the final model of the mediation analysis on filial piety in both sexes after controlling for possible covariates. In older men, FEV1 was inversely correlated with depressive symptom (β = -0.1281, p = 0.004) with no mediation effect of FPE. In older women, FEV1 was negatively associated with FPE, but FPE did not increase the depressive symptom (β = 0.0605, p = 0.12). In both older men and women, FEV1 was negatively associated with PRFP, while PRFP reduced depressive symptom (p < 0.001; Table 3). In older women, the correlation between FEV1 and depressive symptom became insignificant with the mediation of PRFP. In other words, the

**Table 1. Comparison of piety between men and women.**

|  | All | Men | Women | P Value<sup>©</sup> |
|---|---|---|---|---|
| N | 1462 | 727 | 735 |  |
| Age | 74.0±7.6 | 74.6±8.0 | 73.5±7.1 | 0.006 |
| ≥65 | 1261 (86.3) | 627 (86.2) | 634 (86.3) | 0.994 |
| FEV1 | 1.87±0.60 | 2.16±0.62 | 1.58±0.41 | <0.001 |
| Filial piety expectation | 18.06±5.20 | 17.33±5.05 | 18.77±5.24 | <0.001 |
| Respect | 3.38±1.14 | 3.26±1.15 | 3.49±1.12 | <0.001 |
| Make happy | 3.31±1.16 | 3.20±1.18 | 3.42±1.13 | 0.001 |
| Care | 2.70±1.39 | 2.58±1.35 | 2.81±1.42 | 0.003 |
| Greet | 3.31±1.16 | 3.21±1.19 | 3.41±1.12 | 0.001 |
| Obey | 3.21±1.20 | 3.13±1.21 | 3.28±1.19 | 0.026 |
| Financial support | 2.02±1.26 | 1.84±1.16 | 2.20±1.32 | <0.001 |
| Perceived receipt of filial piety | 19.90±4.87 | 19.41±4.78 | 20.38±4.92 | <0.001 |
| Respect | 3.89±1.04 | 3.89±1.08 | 3.90±1.01 | 0.839 |
| Make happy | 3.42±1.07 | 3.33±1.10 | 3.52±1.03 | 0.001 |
| Care | 3.07±1.22 | 2.96±1.24 | 3.18±1.19 | 0.001 |
| Greet | 3.61±1.04 | 3.52±1.06 | 3.70±1.00 | 0.002 |
| Obey | 3.43±1.04 | 3.43±1.03 | 3.42±1.05 | 0.767 |
| Financial support | 2.43±1.20 | 2.24±1.13 | 2.63±1.23 | <0.001 |

relationship between FEV1 and depressive symptom was completely mediated by PRFP in older women. Sensitivity analysis was carried out on subgroups, such as age 65 and above, or with COPD only. Similar results were obtained (Table 3).

## Discussion

This is a largest study to assess the gender difference of mediation effect of filial piety on the relationship between lung function and depressive symptoms in a community-dwelling elderly population. This research divided filial piety into expectation and perceived receipt, and

**Table 2. Association between lung function and depression scale mediated by filial piety.**

|  | All | Men | Women |
|---|---|---|---|
|  | <sup>†</sup>β (95% CI) | <sup>†</sup>β (95% CI) | <sup>†</sup>β (95% CI) |
| FPE |  |  |  |
| Total effect | -0.73 (-1.25 to -0.21) | -1.11 (-1.86 to -0.36) | -0.41 (-1.75 to 0.93) |
| Natural direct effect | -0.72 (-1.25 to -0.19) | -1.11 (-1.86 to -0.36) | -0.40 (-1.75 to 0.94) |
| Natural indirect effect | -0.01 (-0.06 to 0.03) | -0.00 (-0.04 to 0.04) | -0.01 (-0.12 to 0.10) |
| Percentage due to interaction | -1.38 (-16.2 to 13.4) | -0.06 (-5.48 to 5.36) | -0.61 (-54.7 to 52.3) |
| PRFP |  |  |  |
| Total effect | -0.76 (-1.22 to -0.29) | -1.12 (-1.86 to -0.41) | -0.50 (-1.79 to 0.79) |
| Natural direct effect | -0.87 (-1.34 to -0.41) | -1.15 (-1.86 to -0.43) | -0.60 (-1.79 to 0.59) |
| Natural indirect effect | 0.12 (0.02 to 0.21) | 0.03 (-0.05 to 0.10) | 0.10 (-0.23 to 0.43) |
| Percentage due to interaction | 3.58 (-7.00 to 14.2) | 1.12 (-7.96 to 10.2) | -1.48 (-66.9 to 55.0) |

†Adjusted for age, waist, education, smoking status, hypertension, stroke, and MMSE

All analyses were estimated using bootstrapping (5000 replications)

If the mediated effect had a different sign than the other direct effect in a model, the absolute values of the direct and indirect effects were considered prior to calculating the proportion mediated.

A. FPE

B. PRFP

*Men*

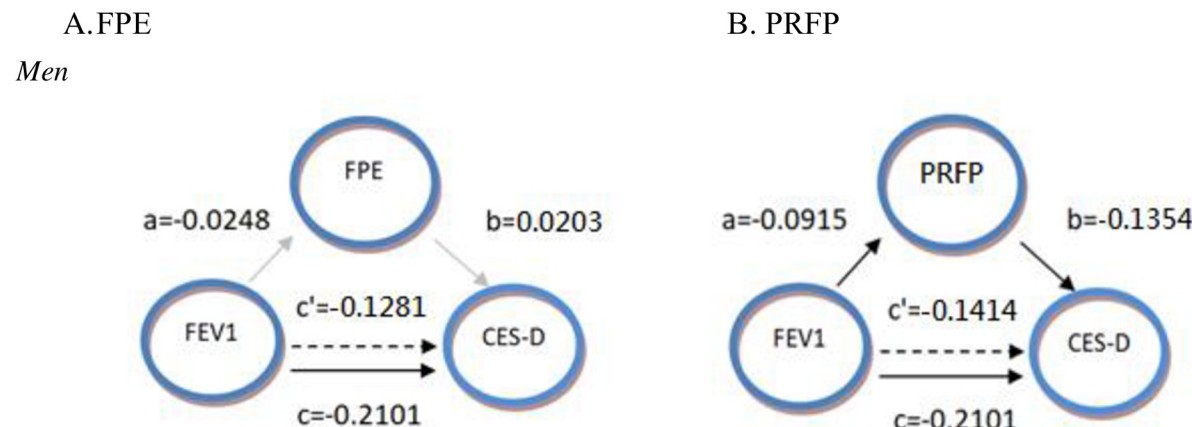

*Women*

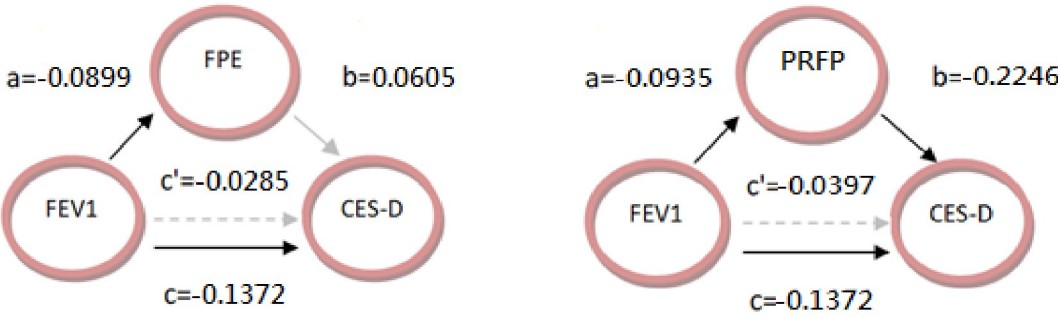

**Fig 1. Mediating model of FPE and PFPR.** Black line: statistically significant. Gray line: statistically insignificant. c was the original relationship between FEV1 and CES-D. c' was the relationship after adding mediator and adjusting for covariates. FEV1: forced expiratory volume in 1 second. CESD: Center for Epidemiologic Studies Depression Scale. FPE: filial piety expectation. PRFP: perceived receipt of filial piety.

investigated their link with worse lung function and depressive symptoms. Our results suggested that PRFP independently modified the association of FEV1 and depressive symptoms in both sexes. After controlling for possible covariates, PRFP was negatively correlated with depressive symptoms. In older men, PRFP had a partial mediation effect on the relationship between FEV1 and depressive symptoms, whereas PRFP had a complete mediation effect on older women. These results indicate that the link between worse lung function and depressive symptoms can be largely mediated by PRFP, especially in older women.

### Association between worse lung function and depressive symptoms

This study found the association between worse lung function and depressive symptoms. Systemic inflammation may play an important role in depression [28]. Serum levels of proinflammatory cytokines such as IL-1, IL-6, and tumor necrosis factor alpha are elevated not only in the patients with depression [29–31] but also in COPD patients [32, 33]. These proinflammatory cytokines play a crucial role in the etiology of depression in COPD [34, 35]. In terms of systemic inflammation, Lu et al. [36] found a strong association of depressive symptoms and

**Table 3. Regression coefficients of the different models predicting the depression scale by the pathway from lung function to filial piety.**

| | model 1 | | model 2 | |
|---|---|---|---|---|
| | †β | P | †β | P |
| Men, FEV1 | -0.1281 | 0.004 | -0.1414 | 0.001 |
| FPE | 0.0203 | 0.611 | - | |
| PRFP | - | | -0.1354 | <0.001 |
| F Value | 3.33 | <0.001 | 5.38 | <0.001 |
| Women, FEV1 | -0.0285 | 0.535 | -0.0397 | 0.356 |
| FPE | 0.0605 | 0.120 | - | |
| PRFP | - | | -0.2246 | <0.001 |
| F Value | 4.69 | <0.001 | 8.61 | <0.001 |
| SENSITIVITY ANALYSES | | | | |
| ≥65 years | | | | |
| Men, FEV1 | -0.1224 | 0.010 | -0.1336 | 0.004 |
| FPE | 0.0026 | 0.951 | - | |
| PRFP | - | | -0.1295 | 0.002 |
| F Value | 3.05 | 0.001 | 4.92 | <0.001 |
| Women, FEV1 | -0.0603 | 0.206 | -0.0578 | 0.201 |
| FPE | 0.0691 | 0.099 | - | |
| PRFP | - | | -0.2002 | <0.001 |
| F Value | 5.00 | <0.001 | 7.84 | <0.001 |
| COPD | | | | |
| Men, FEV1 | -0.0386 | 0.664 | -0.0924 | 0.296 |
| FPE | 0.1291 | 0.143 | - | |
| PRFP | - | | -0.1198 | 0.168 |
| F Value | 0.94 | 0.502 | 1.45 | 0.165 |
| Women, FEV1 | -0.0186 | 0.846 | -0.0471 | 0.576 |
| FPE | 0.1096 | 0.173 | - | |
| PRFP | - | | -0.2794 | <0.001 |
| F Value | 1.41 | 0.182 | 2.89 | 0.002 |

†Adjusted for age, waist, education, smoking status, hypertension, stroke, and MMSE

COPD: FEV1 < 70% predicted or FEV1/ FVC < 0.7

pulmonary function in older adults, which appeared to be mediated by proinflammatory cytokines. High IL-6, high C-reactive protein, and depressive symptoms were independently associated with impaired pulmonary function [36]. It is possible that although COPD is characterized by intense local inflammation in the lungs, it often manifests as extra-pulmonary symptoms including weight loss, muscle wasting, osteoporosis, compromised exercise tolerance, and depression [37, 38].

## FPE and PRFP

To the best of our knowledge, no study has determined whether the association between lung function and depressive symptoms is mediated by filial piety from parents' perspectives. Sex difference in terms of the impact of filial piety on the association between lung function and depression was noted. In our study, older men with worse lung function had higher depression symptoms, and PRFP had a partial mediation effect. Thus, feeling the support and assistance of their children considerably helps to alleviate depression symptoms in older men. Similar to our study, a significant and negative relationship between adult children's filial piety levels and

older parents' depressive symptoms after controlling for multiple covariates has been shown [39]. In older women, the effect of worse lung function on depressive symptoms was completely mediated by PRFP. Thus, after a woman loses lung function, receiving positive support and assistance from family and children and PRFP care can help improve the symptoms of depression. PRFP plays a very important role in the process of aging in women. Interestingly, one study reported that older female parents have more RFP than older male adults [40]. However, sex differences are not significant regarding EFP [40].

Piety is a virtue that has existed for centuries in Chinese society. Many parents still expect piety from their children even though the society has been westernized. Studies have shown that a poorer long-term health status is correlated with lower RFP [39–41]. Our study found that in both sexes, higher FEV1 was correlated with RFP, and higher RFP was associated with lower depressive symptoms. Suggesting that an increase in elderly expectation of their children's filial piety was correlated with a reduction in their poor health status. A recent study reported that lower RFP is associated with an increased risk of perceived stress [42, 43]. With increasing age, the chances of being widowed are higher among women because of a longer life expectancy. We assumed that disadvantaged women are more likely to receive support from their adult children [40, 44]. By contrast, men are much more likely to receive help and support from their spouses. Therefore, the perception of filial piety plays a very important role in the process of female aging. Our results demonstrate that attention should be given to the needs of older women for filial piety, and the potential risk of overlooking the needs of filial piety in elderly men with chronic disease.

The current study had several limitations. This study examined a large group of community-dwelling older adults in Taiwan; therefore, our findings cannot be generalized to other populations. For a more reliable exam, pulmonary tests were repeated five times and the results were averaged. Given that this cross-sectional study design was observational, the causal specifications of lung function and depression in this regard were limited. Nevertheless, we used the SAS CAUSALMED procedure, which works within a counterfactual framework for the analysis. A counterfactual outcome is also called potential outcome, defined for scenarios that might be contrary to the factual. In the counterfactual framework for mediation analysis, intervention on the mediator level is used in different hypothetical situations for defining mediation effects. Procedure CAUSALMED in SAS was developed under this assumption. The procedure decomposes the total effect into 4 components, controlled direct effect (CDE), natural direct effect (NED), natural indirect effect (NIE), and pure indirect effect (PIE). There are two-way, three-way decomposition in this procedure. If we carefully identify relevant confounding covariates, we would have valid interpretation of the mediation effects. The results of the estimation of total, direct, indirect effects are listed in Table 2. Since the effects due to interaction was small, the mediation effects can be estimated. Covariates like age, waist, education, smoking status, hypertension, and stroke were controlled in the model. Residual confounding from unmeasured variables related to filial piety was possible and may have contributed to the observed associations. We also conducted sensitivity analysis on people age 65 years and above or people with COPD (FEV1 < 70% or FEV1/ FVC < 0.7). Similar results were obtained (Table 3). It implied the mediation effects was stable on the relationship between lung function and depressive symptoms. Several variables related to filial piety were lacking such, whether or not adult children lived with the parents. Further studies are needed to clarify these points.

Our study also had several strengths. This was the first study to explore the role of filial care in mediating the association between lung function and depression. This community-based study reflects real life situations and may give health practitioners and policy makers to understand filial piety and its potential influences on older adults' health. Our sample size was large and we tried to control for possible confounders such as age, waist circumference, smoking,

and comorbid medical illnesses. However, caution is required in generalizing the results, since filial piety is important in Asia. Filial piety is a form of social support, especially from the family. Further study on the mediation of social support is recommended.

## Conclusion

Our study showed that feelings of insufficient filial piety may further increase depression, especially in elder subjects with COPD. Although modest, the mediation effect of filial piety focused on improving pulmonary function of the older subjects might decrease depressive symptoms. Our findings provide important implications, especially regarding sex, which can guide future research in enhancing elder filial care. Identifying these mechanisms for different sexes is essential for developing interventions to reduce the subsequent decline in health.

## Supporting information

**S1 Table. Comparisons of characteristics between people with non-COPD and COPD.** (DOCX)

## Acknowledgments

This study was sponsored by the National Health Research Institutes (PH-110-SP-01) and Changhua Christian Hospital (CCH grant no. 105-CCH-IRP-001). Data for this current study has not been previously presented orally or by poster at scientific meetings.

## Author Contributions

**Conceptualization:** Cheng-Chen Chang.

**Data curation:** Agnes C. Hsiung, Hsing-Yi Chang.

**Formal analysis:** Kaichi Hsu, Chun-Min Chen, Si-Sheng Huang.

**Investigation:** I-Chien Wu, Chih-Cheng Hsu, Agnes C. Hsiung.

**Methodology:** Kaichi Hsu, Hsing-Yi Chang.

**Project administration:** Agnes C. Hsiung.

**Software:** Kaichi Hsu.

**Supervision:** Hsing-Yi Chang.

**Validation:** Kaichi Hsu, I-Chien Wu, Chih-Cheng Hsu.

**Writing – original draft:** Cheng-Chen Chang.

**Writing – review & editing:** Hsing-Yi Chang.

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
