## [Decision Letter · Decision Letter 0]

23 Aug 2023

PONE-D-23-15611Gender difference on the mediation effects of filial piety on the association between chronic obstructive pulmonary disease and depressive symptoms in older adults: a community-based studyPLOS ONE

Dear Dr. Hsing-Yi Chang,

Thank you for submitting your manuscript to PLOS ONE. After careful consideration, we feel that it has merit but does not fully meet PLOS ONE’s publication criteria as it currently stands. Therefore, we invite you to submit a revised version of the manuscript that addresses the points raised during the review process.

Kind regards,

Kalyana Chakravarthy Bairapareddy, PhD

Academic Editor

PLOS ONE

Journal Requirements:

Part of this study was sponsored by the National Health Research Institutes (PH-110-SP-01). The funding is mainly for data collection. 

Part of this study was sponsored by the National Health Research Institutes (PH-110-SP-01). The funding is mainly for data collection. 

None.

Additional Editor Comments:

Please respond to the reviewer's comments and add the relevant information that is appropriate in the revised manuscript and resubmit.

Reviewers' comments:

Reviewer's Responses to Questions

**Comments to the Author**

1. Is the manuscript technically sound, and do the data support the conclusions?

Reviewer #1: Yes

Reviewer #2: Yes

2. Has the statistical analysis been performed appropriately and rigorously? 

Reviewer #1: Yes

Reviewer #2: Yes

3. Have the authors made all data underlying the findings in their manuscript fully available?

Reviewer #1: Yes

Reviewer #2: Yes

4. Is the manuscript presented in an intelligible fashion and written in standard English?

Reviewer #1: Yes

Reviewer #2: Yes

5. Review Comments to the Author

Reviewer #1: It is a very interesting study and thank you for allowing me to review this particular research article . Agreed that perhaps more studies are required to find the association between filial piety and depression with worsening lung functions in COPD patients.Future studies may be needed to explore this gap between filial piety, FPR and their outcome thereof in COPD patients. There are a lot of psychosocial factors responsible for depression. Other than that there are also some biological factors which may account for depression like dementia, vascular factors, medical comorbidities etc. Were these taken into account as these could affect the outcome of the study?

Other factors which probably could play a very important role in the study would be the education status of the individual, annual income, marital status, living arrangements, comorbidities of the spouse, immigration status. Could you please throw some light on these factors?

Regarding the COPD status of the individuals, how many of them were on LTOT, home BiPAP or CPAP therapy possibly reflecting the more severe status of the disease and thereby assuming that the level of anxiety and depression in these patients would be more? Was there any such correlation?

Thank you and awaiting your response.

Reviewer #2: This is a very interesting paper.

Was there a review of the population's history of mental illness or use of antidepressant treatment? since the patient with mental illness is known to smoke more.

These patients have a Gold classification for their COPD, since the greater the severity of the disease a greater degree of depression they may have.

6. PLOS authors have the option to publish the peer review history of their article (what does this mean?). If published, this will include your full peer review and any attached files.

Reviewer #1: No

Reviewer #2: **Yes: **Lydiana Avila

---

## [Author Response · Author response to Decision Letter 0]

20 Nov 2023

Reviewer #1: It is a very interesting study and thank you for allowing me to review this particular research article. Agreed that perhaps more studies are required to find the association between filial piety and depression with worsening lung functions in COPD patients. Future studies may be needed to explore this gap between filial piety, FPR and their outcome thereof in COPD patients. There are a lot of psychosocial factors responsible for depression. Other than that there are also some biological factors which may account for depression like dementia, vascular factors, medical comorbidities etc. Were these taken into account as these could affect the outcome of the study?

Reply: Thanks for reminding us the other factors associated with depression. We compared the COPD with non-COPD and found that those with COPD had higher proportion of chronic diseases (supplementary Table 1) and lower cognitive function (MMSE). The inflammatory factors were not different between groups. Most of the participants were able to answer the questions. Those with sever dementia were excluded from the interview. We used MMSE to evaluate their cognitive function (supplementary Table 1). They were different between people with COPD and without COPD. However, their scores indicated they were in relatively good cognitive function. The final model adjusted for age, waist, education, smoking status, hypertension, stroke and MMSE. We did not adjust MMSE in previous manuscript. It was added to the model and did not alter the conclusion. Some coefficients were changed, but the relationships remained the same. We revised the coefficients accordingly. We added one sentence in the section of material and method ‘Age, waist, education, smoking status, hypertension, stroke, and MMSE were adjusted in the final model.’ (Lines 169-170). The footnote of related tables also indicated that these variables were adjusted in the model. 

Other factors which probably could play a very important role in the study would be the education status of the individual, annual income, marital status, living arrangements, comorbidities of the spouse, immigration status. Could you please throw some light on these factors?

Reply: As shown in supplementary Table 1, education was significantly different between the non-COPD and COPD groups, whereas the marital status was not. Education level was adjusted in the final model. We did not have information about the comorbidities of the spouse. Immigration status might not as important as in the US, since most of the participants were Chinese living in Taiwan for long-time.

Regarding the COPD status of the individuals, how many of them were on LTOT, home BiPAP or CPAP therapy possibly reflecting the more severe status of the disease and thereby assuming that the level of anxiety and depression in these patients would be more? Was there any such correlation?

Thank you and awaiting your response.

Reply: We did not have the information about the therapy of the COPD patients. We modeled the lung function as a continuous variable, which might catch various status of patients. 

Reviewer #2: This is a very interesting paper. 

Was there a review of the population's history of mental illness or use of antidepressant treatment? since the patient with mental illness is known to smoke more. 

Reply: We only used the questionnaire data for the analysis. There was no information on the use of antidepressant. These were community dwelling older adults. However, smoking status was controlled in the model. 

These patients have a Gold classification for their COPD, since the greater the severity of the disease a greater degree of depression they may have.

Reply: Indeed, the COPD status might affect the depression status. We analyzed the data using both lung function and depressive symptoms as continuous variables instead of categorical variables. That might reflect the relationship between increasing lung function and depressive symptoms.

---

## [Decision Letter · Decision Letter 1]

24 Jan 2024

Gender difference on the mediation effects of filial piety on the association between chronic obstructive pulmonary disease and depressive symptoms in older adults: a community-based study

PONE-D-23-15611R1

Dear Dr. Hsing-Yi Chang,

We’re pleased to inform you that your manuscript has been judged scientifically suitable for publication and will be formally accepted for publication once it meets all outstanding technical requirements.

Kind regards,

Kalyana Chakravarthy Bairapareddy, PhD

Academic Editor

PLOS ONE

Additional Editor Comments (optional):

Reviewers' comments:

Reviewer's Responses to Questions

**Comments to the Author**

1. If the authors have adequately addressed your comments raised in a previous round of review and you feel that this manuscript is now acceptable for publication, you may indicate that here to bypass the “Comments to the Author” section, enter your conflict of interest statement in the “Confidential to Editor” section, and submit your "Accept" recommendation.

Reviewer #2: All comments have been addressed

2. Is the manuscript technically sound, and do the data support the conclusions?

Reviewer #2: Yes

3. Has the statistical analysis been performed appropriately and rigorously? 

Reviewer #2: Yes

4. Have the authors made all data underlying the findings in their manuscript fully available?

Reviewer #2: Yes

5. Is the manuscript presented in an intelligible fashion and written in standard English?

Reviewer #2: Yes

6. Review Comments to the Author

Reviewer #2: (No Response)

7. PLOS authors have the option to publish the peer review history of their article (what does this mean?). If published, this will include your full peer review and any attached files.

---

## [Editor Report · Acceptance letter]

13 Feb 2024

PONE-D-23-15611R1 

PLOS ONE

Dear Dr. Chang, 

I'm pleased to inform you that your manuscript has been deemed suitable for publication in PLOS ONE. Congratulations! Your manuscript is now being handed over to our production team.

Kind regards, 

on behalf of

Dr. Kalyana Chakravarthy Bairapareddy 

Academic Editor

PLOS ONE